# Rapid In-Process Measurement of Live Virus Vaccine Potency Using Laser Force Cytology: Paving the Way for Rapid Vaccine Development

**DOI:** 10.3390/vaccines10101589

**Published:** 2022-09-22

**Authors:** Reilly McCracken, Noor Al-Nazal, Travis Whitmer, Sijia Yi, James M. Wagner, Colin G. Hebert, Matthew J. Lowry, Peter R. Hayes, James W. Schneider, Todd M. Przybycien, Malini Mukherjee

**Affiliations:** 1Global Vaccines and Biologics Commercialization, Merck & Co., Inc., West Point, PA 19486, USA; 2Vaccines Process Development, Merck & Co., Inc., West Point, PA 19486, USA; 3LumaCyte, Inc., 1145 River Road, Suite 16, Charlottesville, VA 22901, USA; 4Department of Chemical Engineering, Carnegie Mellon University, Pittsburgh, PA 15213, USA; 5Department of Chemical and Biological Engineering, Rensselaer Polytechnic Institute, Troy, NY 12180, USA; 6Analytical Research & Development, Merck & Co., Inc., West Point, PA 19486, USA

**Keywords:** microcarriers, laser force cytology, live virus vaccines, manufacturing, viral potency

## Abstract

Vaccinations to prevent infectious diseases are given to target the body’s innate and adaptive immune systems. In most cases, the potency of a live virus vaccine (LVV) is the most critical measurement of efficacy, though in some cases the quantity of surface antigen on the virus is an equally critical quality attribute. Existing methods to measure the potency of viruses include plaque and TCID50 assays, both of which have very long lead times and cannot provide real time information on the quality of the vaccine during large-scale manufacturing. Here, we report the evaluation of LumaCyte’s Radiance Laser Force Cytology platform as a new way to measure the potency of LVVs in upstream biomanufacturing process in real time and compare this to traditional TCID50 potency. We also assess this new platform as a way to detect adventitious agents, which is a regulatory expectation for the release of commercial vaccines. In both applications, we report the ability to obtain expedited and relevant potency information with strong correlation to release potency methods. Together, our data propose the application of Laser Force Cytology as a valuable process analytical technology (PAT) for the timely measurement of critical quality attributes of LVVs.

## 1. Introduction

The race to vaccinate the world against SARS-CoV-2 and the emergence of several mRNA vaccines has highlighted the need to expedite manufacturing of more traditional vaccines such as live virus vaccines (LVVs). One way to achieve this is by improving control strategies around large-scale manufacturing operations. Unlike some of the newer mRNA vaccines with better molecular characterization of degradation of their critical quality attributes (CQAs) and supporting analytics that can monitor and course correct these changes during manufacturing [1], LVVs have poorly understood mechanisms by which virus potency can be lost [2,3,4,5,6]. However, due to the significant amount of early development and characterization work that goes into manufacturing LVVs, the process definition for these vaccines is usually well-established and tightly controlled. Thus, the simple monitoring of parameters such as cell viability and cell metabolism are relied on as key process parameters (KPPs) to exert reliable control over the process. However, this control strategy often does not fully de-risk the production of virus particles and the monitoring of CQAs such as virus potency and antigenicity. The assessment of these critical attributes relies on classical cell-based potency assays, such as tissue culture infectious dose 50% (TCID50) and plaque assays, which can take as long as 3 weeks and are often manually conducted, leaving room for large variability [7,8,9]. The lack of analytical tools to directly monitor and measure LVV potency rapidly results in a longer iteration time for process development and process characterization of these products, and this is evidenced by the significantly longer times to launch LVVs against infectious diseases (reviewed by [10]). Process analytical technology (PAT) to better track LVVs in real time and ensure their quantity as well as quality has thus become the focus of many big pharmaceutical manufacturers, and PAT is increasingly being recommended by regulatory agencies [8,9]. 

Methods for the direct quantitation of viruses such as detection of virus proteins via enzyme-linked immunosorbent assays (ELISA), quantitation of virus nucleic acid by quantitative reverse-transcription polymerase chain reaction (qRT-PCR), and measurement of viral protein or nucleic acid content by capillary electrophoresis (CE) [11,12] have been used as surrogate measures for potency. However, these analytical methods can only be predictive of potency but will not provide a real measurement of virus potency, which requires infecting cells and evaluating the outcome of a successful infection through measurement of a plaque, cytopathic effect, or virus protein by antibody staining, etc. 

Laser force cytology (LFC) is a unique label-free technology to monitor virus potency by making cell-based optical and hydrodynamic force measurements [13]. Optical and hydrodynamic forces have been used in the past to measure numerous cellular parameters including cell type, viral infection, phagocytosis and cell deformability [14,15,16,17]. LFC captures multiple parameters on a single cell basis, including optical force, size, shape, and deformability without the use of labels or antibodies. The method is attractive as a way to obtain a unique surrogate potency measurement by looking at cellular “fingerprint” upon virus infection in a label-free manner without need for expensive reagents such as antibodies or dyes. Since application of this method is based purely on a unique combination of cellular responses to infection, if successful, one can foresee very broad utility of this technique to monitor potency of vaccines “at-line” not only to manufacture new and future products, but also to assist in the manufacturing of already commercial products such as the M-M-R^®^II, Varivax and ERVEBO^®^ vaccines. Additionally, it is conceivable that this method can be applied to monitor potency of virus-like particles (VLPs), mRNA vaccines and monoclonal antibodies, by measuring the cellular responses induced by expression of each of these entities. The dye-free nature of this assay makes its application relatively easy and dependent only on identifying key cellular features that are altered by introduction of foreign agents into cells. 

In this study, we present the application of LFC as a valuable PAT to monitor the estimated potency of LVVs in upstream bio manufacturing processes with the goal of improving the speed, efficiency and quality of vaccine development and manufacturing. First, using Vero cells grown on microcarriers and infected with a Measles GFP virus, we establish the cellular parameters that display a most consistent and robust change upon virus infection compared to uninfected cells. After establishing this unique cellular fingerprint, we then evaluate correlation of this to an already established method to measure Measles virus particle counts using flow virometry. We find that an excellent correlation exists between total particle counts and virus potency measured by Lumacyte. Next we evaluate the ability of this method to distinguish between two different viruses infecting the same production bioreactor, with the objective of monitoring and detecting adventitious agents, which is a regulatory requirement in the manufacturing of vaccines. In this goal, we now look at the differences in the cellular fingerprints of the two viruses and find unique and non-overlapping patterns of detection that can be used to separate an adventitious infection from the real production culture. Finally, and in the last section of this study, we also evaluate the application of Lumacyte’s Radiance instrument as a way to obtain a traditional potency value for the Measles virus to potentially replace or supplement a TCID50 assay. Using the same cellular fingerprint already established for this virus, we find that Radiance is able to predict a potency value for the virus which is highly correlated to the TCID50 value and has a much faster turnaround time (2–3 days for Radiance versus 6–8 days for TCID50).

## 2. Results

### 2.1. Live Virus Microcarrier Culture Bioreactor Monitoring Current Practice

As shown in Figure 1, the manufacturing of LVVs involves several analytical tools that are used at various stages throughout the process to measure multiple parameters including cell growth and viability. The ability to monitor the production of virus in real time, as opposed to waiting several days for a typical infectivity assay, would allow for rapid process optimization, increase process knowledge, uncover sources of variability, and potentially increase product yield by more precisely timing the harvest. While LFC could potentially be applied to several stages of the upstream process, this study focuses on the Virus production phase in order to determine the ability of the LumaCyte Radiance instrument to provide cell-based measurements that could be used to calculate viral production throughout the time course. 

### 2.2. Laser Force Cytology Cellular Fingerprinting of Measles Virus Infection Correlates with Total Virus Particle Counts and Acts as a Surrogate to Potency

Figure 2 provides a detail of the sample harvest process for samples run on Radiance. Initially, Vero cells were seeded onto microcarriers and then incubated for a period of time to allow for the cells to become confluent, at which point they were infected with attenuated MV. At each timepoint post infection, a sample was withdrawn from the bioreactor and separated into two fractions. The first contains the microcarriers with cells attached, and the second contains the supernatant as well as any cells that have separated from the microcarriers throughout the production process. The microcarrier fraction is then processed using an enzymatic solution and several wash steps to detach the cells as described further in Section 3. From there, the separate microcarrier and supernatant samples are centrifuged and then resuspended in LumaCyte Stabilization Fluid at a concentration of approximately 750,000 cells/mL. A volume of 200 µL (150,000 cells total) is then transferred to a 96-well plate for analysis with Radiance as separate samples. 

As demonstrated previously [13,18], LFC can be used to monitor the infection time course by identifying one or more parameters measured by Radiance that can be correlated with the viral infectivity, referred to as the Infection Metric. In the case of other viruses [13,14], the optical force, and thus velocity, as well as the size changes with infection and can be incorporated into the infection metric. Therefore, an initial starting point for determining the cellular changes that occur throughout the infection time course is the Size vs. Velocity scatter plot. A scatter plot data from a 3 L bioreactor experiment is presented in Figure 2B. Specifically, each graph compares the uninfected cells (green) detached from the microcarriers just prior to infection to the supernatant sample collected at the time indicated on the graph (red), starting with 3 days post infection. In general, infection levels are below limit of detection prior to day 3 post infection across multiple measurements including LFC and other potency assays. 

As can be seen in Figure 2B, as the infection progresses from 3 to 6 days post infection, the infected supernatant population shifts substantially away from the uninfected population, overall decreasing in velocity for a given size, reflecting an increase in the optical force index (OFI) or size normalized velocity. As discussed previously, OFI was developed to help normalize any size changes that occur with infection and help describe the effects on optical force independent of cell size [13]. While a decrease in velocity represents an increase in optical force, the velocity component of the OFI parameter is the difference between the fluid velocity and the cell velocity. Thus, an increase in OFI represents an increase in the optical force exerted on the cell. OFI histograms for the supernatant fraction of the same dataset are presented in Figure 2C and show a progression of increasing optical force as the infection progresses. Similar to gating in flow cytometry [19], the percentage of a population above a threshold value can be used to help describe the population. In this case, the empirically determined threshold of 55 (s^−1^) was selected and is shown on each of the histogram plots. As shown in Figure 2C, the percentage of cells above 55 increases from 23.6% at 3 days post infection to 71.5% at 6 days post infection, demonstrating an increase in the optical force as the infection progresses. Based on this behavior, the OFI threshold value was selected as the infection metric. 

Once identified, the infection metric can be used to track the progress and kinetics of an infection. However, to use LFC to calculate other biological parameters, such as LVV potency, a quantitative correlation must be developed between the biological parameter of interest and the infection metric. In this case, the biological parameter used was total virus particles in the supernatant, as measured by CytoFLEX and presented in Figure 2E. Previously, this has been shown to provide a good correlation to the potency for MV, which is a TCID50 assay. Based on these previously established data showing consistent total to infectious particle ratio using CytoFLEX and TCID50 assays for the Measles virus, we have used total particles counts generated by CytoFLEX as a “surrogate” potency measurement for this study. As can be seen in Figure 2E, the amount of total virus particles increases throughout the infection process, and there is some variability in productivity between the 3 independent bioreactor runs analyzed for this dataset. 

Because LFC makes measurements on a *per-cell* basis and the viral particles measurement in this case is a *per-volume* measurement, the cell count must also be considered in order to accurately correlate the LFC data with the viral particle measurement. Cell counts, both for the microcarriers and supernatant fractions, are shown in Figure 2D for each of the three experiments. As illustrated in the figure, no significant detachment of cells from the microcarriers into the supernatant is seen until 3 days post infection, after which point the percentage of detached cells increases significantly. Based on the cell count of both the microcarriers and supernatant fraction as well as the number of virus particles per mL, the number of virus particles per viable cell can be calculated and is shown on the *Y*-axis of Figure 2G. As Radiance measures the microcarriers and supernatant fractions separately, the infection metric also needs to be adjusted based on the overall percentage of the cell population represented by the microcarriers and supernatant fraction, respectively. Specifically, the population combined OFI metric was calculated based on the following equation:$$Combined\;OFI\;Metric\;\left(\%\right)={\frac{{\frac{cells}{mL}}_{M}}{{\frac{cells}{mL}}_{Total}}}_{\;}\times \%\;cells\;OFI>55_{M}+\frac{{\frac{cells}{mL}}_{S}}{{\frac{cells}{mL}}_{Total}}\times \%cells\;OFI>55_{S}\;$$

The combined OFI metric taking into account both the microcarriers and supernatant fraction is presented in Figure 2F and is the X parameter of the scatter plot in Figure 2G, showing a strong correlation with the total virus particles per viable cell. The log_10_ difference between the measured and calculated values are shown for each time point in Table 1, with an absolute average difference of 0.074 log_10_. This demonstrates the capability to dynamically monitor the estimated yield of virus using the Radiance data, allowing for a rapid measurement of estimated potency that could be used to understand, monitor, and potentially adjust the production process in minutes as opposed to hours or days for typical potency measurements. 

### 2.3. A Unique and Non-Overlapping Fingerprint of the Adventitious Virus Is Established in a Microcarrier Bioreactor Culture and Correlated to Total Virus Particle Counts

In addition to MV, another virus of the Flavivirus genus, designated AV1, was also investigated in this study as a simulated adventitious virus. Similar to MV, AV1 was infected onto Vero cells planted onto microcarriers and then sampled periodically for analysis with Radiance across 3 independent bioreactor runs performed at different times. Single cell results from one of the experiments is presented in Figure 3A,B. Size vs. velocity scatter plots presented in Figure 3A compare uninfected cells (green) detached from the microcarriers just prior to infection to the supernatant sample collected at the time indicated on the graph (red), starting with 3 days post infection. Similar to what was seen observed with MV infection, the infected supernatant population decreases in velocity compared to the population prior to infection, and the magnitude of the velocity change increases as the infection progresses. This results in an increase in the OFI of the population, as illustrated in Figure 3B. The percentage of cells with an OFI > 55 s^−1^ is shown on each histogram, increasing from 34.4% at 3 days post infection to 77.7% at 6 days post infection. 

In order to determine the potential for using LFC measurements to monitor the production of AV1, cell count and viral particle counts were collected for each time point and are presented in Figure 3C,D, respectively. As was seen with MV, supernatant cells that have spontaneously detached from the microcarriers start to appear at 3 days post infection and increase significantly at 4 days post infection. Similarly, the total number of viral particles also increases throughout the infection. The combined OFI metric, which takes into account both the microcarriers and supernatant fractions, is shown in Figure 3E, and the correlation between this OFI metric and the total viral particles per viable cell is shown in Figure 3F. The log_10_ difference between the measured and calculated values are shown for each time point in Table 2, with an absolute average difference of 0.100 log_10_. Similar to MV, there is a strong correlation between the OFI metric and the total viral particles/cell, demonstrating the capability to monitor the estimated potency of AV1. Showing this capability for multiple viruses demonstrates the broad applicability of LFC to measure viral infection and develop a correlation with estimated potency. To date, LFC has been tested with over 30 cell lines and primary cell types as well as over 35 different viruses (data not shown). 

### 2.4. The Unique Cellular Responses Measured by Laser Force Cytology Are Used to Identify and Distinguish between Measles and an Adventitious Virus in a Co-Infected Production Bioreactor

The detection of adventitious agents, including viruses, is an important component of biopharmaceutical production. Ideally, products could be screened throughout the manufacturing and production process, enabling real-time or rapid detection of any potential contamination in order to take action as soon as possible. In the case of LVVs, this process is sometimes complicated by the fact that viruses are used to help produce the product, and thus any adventitious virus must be detected in the background of an ongoing virus infection. In order to test the capability of Radiance to differentiate between multiple viruses and combinations, an experiment was conducted using a bioreactor coinfected with the MV virus (1300 pfu/cm^2^) as well as AV1 (130 pfu/cm^2^) and supernatant and microcarrier samples were again monitored throughout the production process. In addition to the coinfection condition, individual virus controls of the MV (1300 pfu/cm^2^) as well as AV1 (1300 pfu/cm^2^) were monitored in parallel. Results are shown in Figure 4. Figure 4C shows the average OFI of the microcarrier and supernatant fractions for each reactor condition throughout the time course. As with previous experiments, a significant number of supernatant cells was not detected until 72 h post infection, so only microcarrier samples are shown up until that timepoint. In general, minimal changes are seen 24 h and 48 h post infection, and there are no significant changes in OFI between the three conditions. At 72 h, there is an increase in the OFI of the microcarrier and supernatant AV1 samples compared to the MV and MV + AV1 conditions. However, there is no differentiation of the MV + AV1 condition from the MV alone based on optical force index, as they are statistically similar in both the microcarrier and supernatant fractions. However, this changes at the later timepoints. At the 96 h timepoint, there is a further increase in the OFI of the AV1 samples compared to the other two conditions in both the microcarrier and supernatant fractions, with the AV1 samples showing clear differentiation (*p* < 0.01) between the MV and MV + AV1 samples. There is also a slight increase in the OFI of the microcarrier fraction of the MV + AV1 sample versus the MV sample but based on OFI alone the difference is not statistically significant. At the 168 h timepoint, there is a larger and statistically significant (*p* < 0.05) difference between the microcarrier fraction of the MV and MV + AV1 conditions, as potentially the longer timepoint has allowed for an increased influence of the AV1 relative to MV. 

Additional detail and information can be gathered by using the single cell data collected using LFC. Representative histograms for the microcarrier (Figure 4A) and supernatant (Figure 4B) fractions at the 96 h timepoint are shown for each of the three sample types and illustrate the potential for using the population distribution to differentiate between them. For example, when looking at the MV and AV1 microcarrier histograms, there is a distinct shape difference between the two. Specifically, the MV population has a flat profile with very few cells having an OFI > 50 s^−1^, while the AV1 population has a large peak at an OFI of approximately 40 s^−1^ that tails off and includes a comparatively larger number of high OFI cells. The MV + AV1 coinfection population is a blend of the two, having a substantial number of low OFI cells similar to the MV single infection as well as having the peak at higher OFI similar to the AV1 single infection. Differences are seen between the supernatant histograms as well, as shown in Figure 4B. The MV population is composed predominantly of low OFI cells, showing a high peak around OFI 10 that declines consistently and has few high OFI cells. In contrast, the AV1 supernatant sample has a comparatively smaller population of low OFI cells and instead has a broadly distributed population with a larger standard deviation. 

Another approach afforded by LFC can use multivariate statistical methods to include additional population-wide parameters in addition to the OFI. In this case, principal component analysis (PCA) using the average and standard deviation of 21 parameters measured by Radiance was used to classify the microcarrier and supernatant samples at the 96 h timepoint (Figure 4D). PCA is mathematical technique used for multivariate data that refactors multiple parameters into principal components (PC1 and PC2 shown on the graph) to allow for improved visualized of multivariate data and capture the greatest amount of variance in the data [20]. In this case, it can be used to group and characterize the microcarrier and supernatant samples for each of the three different conditions. Each point on the graph represents a separate well from the same initial sample analyzed using Radiance. Several distinctions can be made using the PCA plot. First, there is a clear separation between the microcarriers samples and the supernatant samples. In addition, there is separation between the AV1 samples and the other two conditions, with a larger separation seen in the supernatant samples. The coinfection samples for both the microcarriers and supernatant samples are located between the other two conditions but closer to the MV samples, which likely follows from the initial MV:AV1 ratio of 10:1. 

These initial results it illustrates the potential of Radiance to detect an adventitious virus in a viral production background. With the collection of additional data, the single cell and multivariate analysis techniques can be refined and expanded to increase process knowledge. Once a process is well understood, differences seen using the PCA plot or other metrics could be used to rapidly identify any process changes that could indicate adventitious agents or other variations that would warrant further attention. 

### 2.5. Measurement of Measles Virus Potency in a Plate-Based Assay by Laser Force Cytology Shows Robust Correlation to the Traditional TCID50 Assay

Infectivity assays were conducted as initial tests to determine cell properties that change with measles infection and whether different levels of initial virus inoculations can be quantified using Radiance. Cells were infected at several initial MOIs then harvested and measured in Radiance at 1, 2, and 3 DPI (Figure 5A). Changes were observed as early as 1 day post infection at the higher MOIs with differences observed at all MOIs by day 3 post infection. Comparing both the size and velocity of cells at 3 DPI, infected samples had a subpopulation of cells that were smaller in size and had a lower average velocity compared to the uninfected population (Figure 5B). The number of cells within this subpopulation of smaller size and lower velocity cells appears to change with the initial MOI applied to the sample with the highest MOI having the largest subpopulation. Since both of these properties appeared to correlate with different levels of virus infection, OFI was selected for further analysis. An empirically derived threshold of 35 s^−1^ was applied and the percentage of cells in each sample above this cutoff was calculated (Figure 5C). At 3 DPI, the highest MOI tested of 4.1 had 30.5% of cells above the optical force index threshold while the lowest MOI of 0.013 had only 5.3% above. Two independent infectivity assays were conducted with Vero cells from different cell banks and two virus stocks of different titers. In both experiments, the Radiance infection metric (percentage of cells with optical force index greater than 35 s^−1^) was observed to highly correlate to the initial inoculum level of virus using a power law fit as seen by the R^2^ values of 0.99 (Figure 5D). The log_10_ differences between the measured and calculated values for both curves are shown in Table 3, with an absolute difference of 0.024 log_10_ and 0.001 log_10,_ respectively, when considering the average across all the measured dilutions. This shows the utility of Radiance for accurately determining titers of measles virus samples.

## 3. Methods

### 3.1. Bioreactor Sampling

After a period of growth, cells were infected with MV or AV1 at a defined MOI and monitored for up to 6 days post infection. Samples containing both microcarriers and supernatant were withdrawn daily for cell count, viability, viral particle, and Radiance analysis. For Radiance analysis, microcarriers with attached cells were separated from any free-floating cells in the supernatant by allowing the microcarriers to settle for about 1 min and then transferring the supernatant to a separate tube. In order to detach the cells from the microcarriers, samples were rinsed with two separate volumes of TrypLE (Thermo Fisher Scientific, Waltham, MA, USA) prior to a 10 min incubation time in TrypLE at 37 °C to detach the cells. Following this incubation, samples were vigorously mixed to help detach the cells from the microcarriers. Both the microcarriers and supernatant fractions (still separate) were then filtered using a 37 m cell strainer to remove any remaining microcarriers. Both fractions were then run on a Vi-Cell Blu Cell Viability Analyzer (Beckman Coulter, Brea, CA, USA) for determination of viable cell density. Samples were then centrifuged for 5 min at a speed 200 g prior to resuspension LumaCyte Stabilization Fluid with 0.5% paraformaldehyde (PFA) to a final concentration of approximately 750,000 cells/mL. Cell samples (200 µL) were then loaded into the 96-well plate for analysis with a Radiance instrument.

### 3.2. Cell Culture

Vero cells (ATCC CCL-81) were cultured in Dulbecco’s Modified Eagle’s Medium (DMEM) growth medium (Thermo Fisher) supplemented with 10% heat-treated FBS (Thermo Fisher) and 1% Penicillin-Streptomycin (Pen/Strep) (Thermo Fisher) in a humidified incubator at 37 °C and 5% CO_2_ and passaged using TrypLE. Cell samples were counted using a Vi-CELL BLU Cell Viability Analyzer for determination of viable cell density.

### 3.3. Viral Infectivity Assay

Vero cells were seeded into 24-well plates at a density of 200,000 cells/well the day prior to infection. On the day of infection, a measles virus stock of known titer was thawed and diluted down a 5-fold dilution series in DMEM assay medium supplemented with 2% FBS and 1% Pen/Strep. The growth media was aspirated from all wells and 200 µL of the appropriate virus dilution was added to each infected condition well to achieve the various MOIs tested (2, 1, 0.2, 0.04, 0.008, 0.0016). Cells were incubated for 4 h when an additional 200 µL of fresh assay medium was added to each well. Samples were further incubated until the collection time points at either 1, 2, or 3 DPI.

### 3.4. Sample Collection for Infectivity Assay

Cells were washed with PBS and harvested using TrypLE. Cells were then washed with assay medium and centrifuged at 200× *g* for 5 min. Cells were re-suspended into stabilization fluid (LumaCyte, Charlottesville, VA, USA) with a final concentration of 0.5% paraformaldehyde (Electron Microscopy Sciences) to achieve a concentration of approximately 500,000 cells/mL. Then, 200 µL of each cell sample was used for measurement on Radiance (LumaCyte).

### 3.5. CytoFLEX Assay

The CytoFLEX (Beckman Coulter) instrument used has three laser colors and assorted detectors, with violet side scatter (VSSC) being the laser of relevance for this study. NERL^TM^ Reagent Grade water from ThermoFisher Scientific was used as the sheath fluid for all experiments. Prior to running samples, Quality Control beads (Beckman Coulter) were run on the instrument. Samples were diluted in 0.1 µm filtered PBS and then run on the instrument at a 10 µL/min flow rate. Gating for virus particles was performed using the CytExpert (Beckman Coulter) software. Virus particle concentrations for each sample were determined by multiplying the measured events/µL by the dilution factor and the percentage of particles within the gated area.

### 3.6. Data Analysis

LumaCyte Radiance data were analyzed using LumaCyte’s ReportR software. Flow cytometry data were analyzed using CytExpert software (Beckman Coulter). All graphs were generated using Microsoft Excel. The power regression equations and coefficients of determination (R^2^) for correlations were determined using Microsoft Excel.

## 4. Discussion

The recent COVID-19 pandemic has highlighted the need to improve the speed of the development and manufacturing of vaccines of all types. LVVs have an established history of safety and long-term efficacy, but their development has been comparatively slow, often taking decades or more [5,21,22]. One factor contributing to the slow pace of development is the challenge associated with accurately characterizing the potency of LVVs, which historically has been achieved using traditional potency assays such as plaque or TCID50 assay [23,24]. These assays are labor intensive, highly variable, and often subjective [8,9], making them in general a rate limiting step in the process development and process characterization stages of LVV manufacturing. In the commercial space, these methods, though still used as gold standards for release potency measurement, cannot be used for routine monitoring of virus particles in the upstream or downstream space. This leaves room for gaps in control strategy and the potential for costly batch failures due to out of specification potency results. Suitable PAT tools that can be used to quickly and reliably monitor and release batches in real time would have numerous benefits in terms of ensuring the quality, providing opportunity for early intervention and maximizing the quantity of LVVs. This study highlights the use of LFC as a potential tool for providing rapid results and insight for multiple facets of the development and manufacturing process in microcarrier based mammalian cell culture platforms, which are arguably more complex than static cell cultures and require advanced control strategies.

First, the use of LFC as a PAT for monitoring vaccine potency was investigated using two viruses, MV and AV1. In both cases, the infection was tracked throughout multiple independent bioreactor runs using cellular changes measured by LFC in both the supernatant and microcarriers cell fractions. Importantly, by combining the data from both fractions, a strong correlation was developed between the percentage of high optical force cells (OFI > 55 s^−1^) and the estimated potency, with a calculated average absolute log_10_ difference of 0.07 for MV and 0.10 for AV1, respectively. This demonstrates the application of LFC to provide rapid feedback on potency measurements to improve the speed of process optimization by reducing time to result as well as provide ongoing feedback to inform the manufacturing and potentially downstream purification processes by optimizing the purification steps based on the potency readout. LFC is a dye- and label-free assay that measures changes in intrinsic cell parameters upon virus infection, thus reducing the cost and variability associated with reagent based methods. Importantly, most currently available rapid at-line methods of detecting total virus particle counts (like flow virometry used in this report) can provide a biophysical measurement of virus particles but cannot provide a read out on virus quality and infectivity. As infectivity is often the key CQA of interest, such surrogate methods for estimating potency thus require very large datasets to build out robust correlations to a real potency value obtained using either a TCID50 or a plaque assay. LFC represents one of the very few available ways by which an estimate of a true infectious titer can be obtained in real time and if further developed, can significantly advance control strategy in this realm. 

Another critical aspect of LVV as well as other biologics production is the detection of adventitious agents, including adventitious viruses. Typically, screening for adventitious viruses occurs after production process, especially in the case of a virus-based process, since generally the production virus is neutralized or inactivated as part of the screening assay. Having an early indication of a possible contamination during the production batch itself would be advantageous and could minimize the loss of additional time and resources associated with a contamination. In addition, if the speed of the viral screening assay could be improved, this would reduce one potential bottleneck in the process of releasing product. In this study, we show the potential for LFC as an in-process tool for detecting adventitious viruses by detecting differences between individual infections of MV and AV1 as well as co-infection of AV1 into a MV production background. Using a second (adventitious) virus of the Flavivirus genus, we tested another enveloped virus (similar to Measles virus) and hypothesized that a difference in cellular signature between the two viruses would be detected only if the mechanistic differences in virus infection and its effect on the cell were distinct enough to be detected and differentiated by Radiance. Differences were seen starting at 96 h post infection, both in the single cell optical force histograms and using multivariate PCA analysis. While additional studies are warranted, this does illustrate the ability for LFC to rapidly detect subtle differences between cellular populations that can potentially be used to increase process knowledge and sources of variability, with the goal of ultimately increasing product quality and quantity. We also expect a lot of the most common adventitious agents to be from the bacterial, fungal or mycoplasma family with vastly different and distinct infection patterns from that of an LVV and being fairly easily detected by our method. Another area for future study is the investigation of the biological basis for the differences measured between the two viruses and specifically, to examine how LFC could potentially classify the cell-based response to a wide variety of viruses, for example ssRNA vs. dsDNA or enveloped vs. non-enveloped viruses, as well as the stage at which the difference occurs, whether it be entry or replication. Such mechanistic studies could help tease apart differences in infectivity between viruses and other process conditions, potentially leading to the design of more effective vaccines. For example, theoretical studies examining the optimal viral configuration for entry to maximize potency could be investigated [25,26], resulting in a large number of candidate vaccines. Screening of these candidates could be streamlined using LFC as a potency measurement with a faster time to result. 

In addition to measuring potency during the manufacturing process, there is also a need to measure potency during various stages for process and formulations development purposes, as well as in the final product prior to release. Here, we demonstrate a robust correlation between LFC based optical force measurements at 3 days post infection and potency as measured by a TCID50 assay, which takes 6–8 days. Detection and correlation with LFC were also possible for higher titer values at only 2 days post infection, with 3 days providing the maximum range of detection for the MOIs tested. It is important to note that this demonstrates a correlation with an infectious viral titer, rather than a physical titer based on the total number of viral particles. In production, it is likely that the ratio between total and infectious virus particles is generally consistent, but should this ratio change for some reason during the purification or storage process, it is critical to have a potency assay that can account for these differences. 

In summary, the use of LFC as a rapid PAT for monitoring viral infection and estimated potency as well as an analytical assay for measuring infectious titer at-line helps pave the way for reducing the research, development, and manufacturing timeline for LVVs as well as other vaccines that rely on viruses during their manufacturing and development process, such as protein subunit vaccines produced in Sf9 cells via baculovirus or adenovirus-based viral vector vaccines [27]. In general, the capability to make rapid, cell-based infectivity measurements has the potential to improve the entire vaccine development life cycle from R&D to clinical trials and manufacturing, reducing the cost and time associated with LVVs and other vaccines. 

## Figures and Tables

**Figure 1 vaccines-10-01589-f001:**
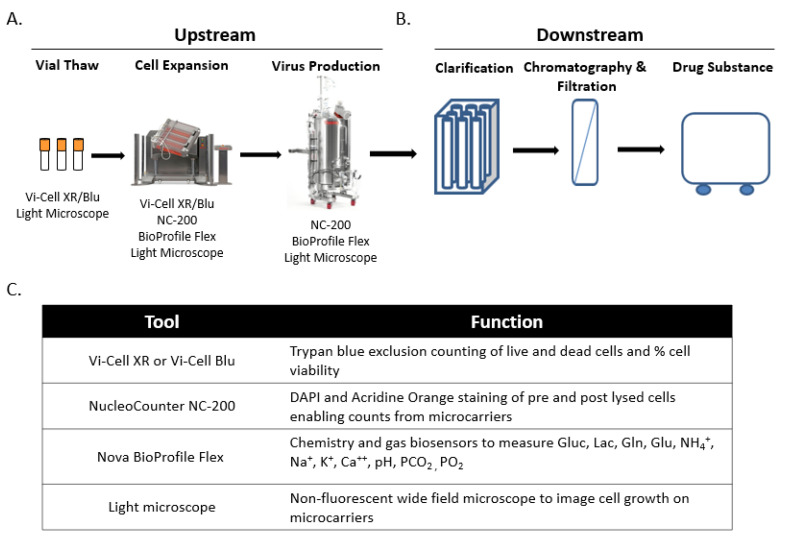
Depiction of a typical microcarrier live virus vaccine manufacturing process and associated in-process analytical tools. (**A**). The upstream LVV process starts with thawing a vial of cells that are planted and expanded in a series of cell stacks. The adherent cells are then taken from the cell stacks and planted in bioreactors on microcarrier beads to expand. During the cell growth stage, Vi-Cell and/or NC-200 counts are used to measure the viable cell concentration and viability. After growing on microcarriers in bioreactors for ~5 days, cells are infected with the virus of interest. During the virus production phase, NC-200 is used to measure viable cell concentration and viability. BioProfile Flex is used to monitor bioreactor pH and metabolites for both the cell expansion and virus production phases, and a light microscope is used as a visual indicator of cell viability and/or attachment to microcarriers throughout the entire upstream process. (**B**). The downstream process starts with obtaining harvested virus from the production bioreactor. The harvested virus is then processed through a series of clarification, chromatography, and filtration steps before being frozen as the final drug substance (DS). This process currently does not utilize any in-process analytical tools to monitor viral potency throughout downstream processing. (**C**). Table of the current in-process analytical tools used throughout the LVV manufacturing process and their functions.

**Figure 2 vaccines-10-01589-f002:**
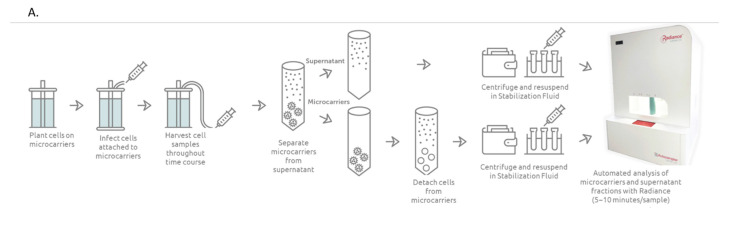
LumaCyte Radiance analysis of Vero cells grown on microcarriers and infected with MV at the 3 L bioreactor scale resulted in an infection metric that correlates with total particle counts. (**A**). Bioreactor and cell preparation workflow for cells grown on microcarriers in 3 L bioreactors. Cells are added to a bioreactor containing microcarriers and grown for five days before infecting with virus. Cell samples are taken throughout multiple days post-infection and separated into supernatant and microcarrier fractions before analyzing on LumaCyte’s Radiance instrument. (**B**). Representative scatter plots compare cell area-based size vs. velocity profiles for uninfected cells (blue) and MV infected cells (red) at 3 through 6 days post-infection (3–6DPI). Each dot represents a single cell analyzed by Radiance. (**C**). Representative histograms show the differences in optical force index profiles for MV infected cells for days 3 through 6 post-infection (3–6DPI). Bar heights represent the number of cells with an optical force index falling within the given bin range (bin width = 2). The dotted line represents the optical force index threshold set as the Radiance infection metric (OFI > 55 s^−1^) for this study. The percentage of the cell population with an OFI > 55 s^−1^ is shown on each plot. (**D**). Viable cell concentration for the supernatant (S) and microcarrier (M) cell fractions of MV-infected cells measured using Vi-CELL BLU at each time point. (**E**). CytoFLEX total virus particle concentration of MV samples from three 3 L experiments across varying days post infection (*n* = 3). (**F**). Progression in Radiance infection metric (percentage of cells with OFI > 55 s^−1^) for cells infected with MV from 0 days post-infection (uninfected) through 6 days post-infection across three 3 L bioreactor experiments. This value accounts for the percentage of cells in both the S and M fractions (*n* = 3). (**G**). Correlation between Radiance infection metric (percentage of cells with OFI > 55 s^−1^) and CytoFLEX total particle counts normalized to the viable cell count for MV infected cells across three independent bioreactor experiments. Each point represents the timepoint and experiment indicated on the plot (R^2^ = 0.92, *n* = 3).

**Figure 3 vaccines-10-01589-f003:**
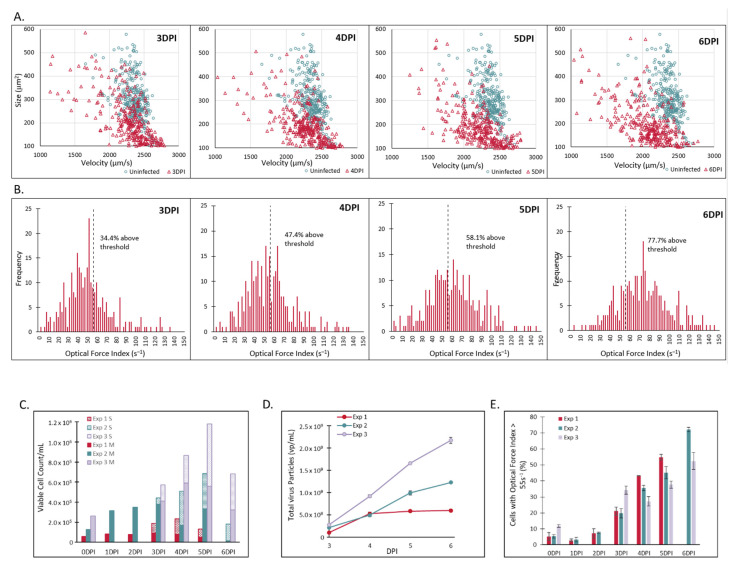
LumaCyte Radiance analysis of Vero cells grown on microcarriers and infected with AV1 at the 3 L bioreactor scale resulted in an infection metric that correlates with total particle counts. (**A**). Representative scatter plots compare cell area-based size vs. velocity profiles for uninfected cells (blue) and AV1 infected cells (red) at 3 through 6 days post-infection (3–6DPI). Each dot represents a single cell analyzed by Radiance. (**B**). Representative histograms show the differences in optical force index profiles for AV1 infected cells for days 3 through 6 post-infection (3–6DPI). Bar heights represent the number of cells with an optical force index falling within the given bin range (bin width = 2). The dotted line represents the optical force index threshold set as the Radiance infection metric (OFI > 55 s^−1^) for this study. The percentage of the cell population with an OFI > 55 s^−1^ is shown on each plot. (**C**). Viable cell concentration for the supernatant (S) and microcarrier (M) cell fractions of AV1-infected cells measured using Vi-CELL BLU at each time point. (**D**). CytoFLEX total virus particle concentration of MV samples from three 3 L experiments across varying days post infection (*n* = 3). (**E**). Progression in Radiance infection metric (percentage of cells with OFI > 55 s^−1^) for cells infected with AV1 from 0 days post-infection (uninfected) through 6 days post-infection across three 3 L bioreactor experiments. This value accounts for the percentage of cells in both the S and M fractions (*n* = 3). (**F**). Correlation between Radiance infection metric (percentage of cells with OFI > 55 s^−1^) and CytoFLEX total particle counts normalized to the viable cell count for AV1 infected cells across three independent bioreactor experiments. Each point represents the timepoint and experiment indicated on the plot (R^2^ = 0.92, *n* = 3).

**Figure 4 vaccines-10-01589-f004:**
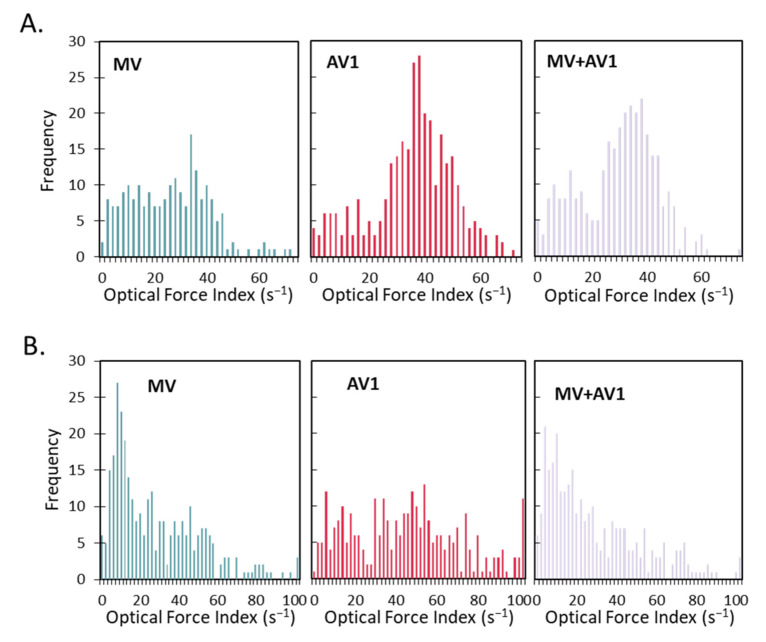
LumaCyte Radiance demonstrated the ability to identify differences in cell samples supplied from 3 L bioreactors infected with MV alone, AV1 alone, and a combination of MV + AV1. (**A**). Representative OFI histograms from microcarriers samples for MV, AV1, and MV + AV1 coinfection at 96 h post infection. (**B**). Representative OFI histograms from supernatant samples for MV, AV1, and MV + AV1 coinfection at 96 h post infection. (**C**). Population average OFI for microcarriers and supernatant samples across the production time course for MV, AV1, and MV + AV1 coinfection (error bars represent technical replicates *n* = 3). (**D**). Principal component analysis for MV, AV1, and MV + AV1 samples using Radiance data. Each point represents a sample analyzed with LFC from either the microcarriers or supernatant samples at 96 h post infection.

**Figure 5 vaccines-10-01589-f005:**
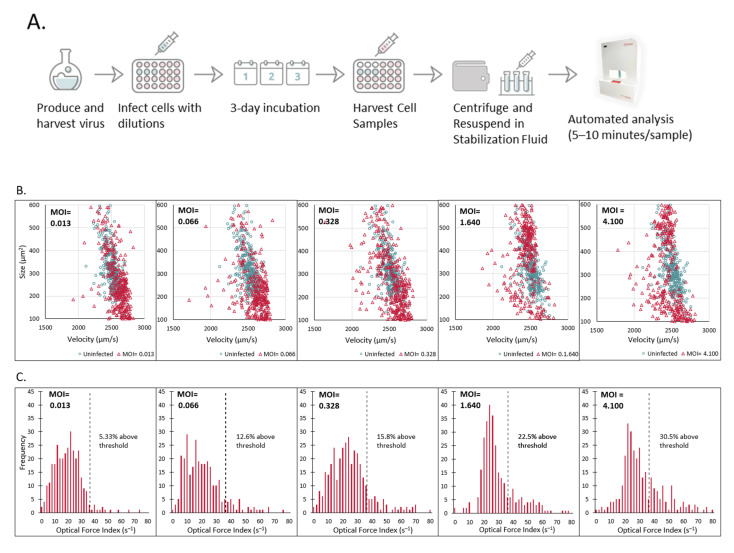
LumaCyte Radiance analysis of Vero cells grown on 24-well plates and infected with MV at varying MOIs resulted in an infection metric that correlates well with MOI. (**A**). Workflow for MV infectivity assay. Cells are grown on 24-well plates and infected with MV at varying MOIs. At 3 days post-infection (3 DPI), cells are harvested from the plates, centrifuged, and resuspended in stabilization buffer before analyzing on LumaCyte’s Radiance instrument. (**B**). Representative scatter plots compare cell area-based size vs. velocity profiles for uninfected cells (blue) and MV infected cells (red) at 3 days post-infection (3 DPI). Each graph represents cells infected with a different MOI. Each dot represents a single cell analyzed by Radiance. (**C**). Representative histograms show the differences in optical force index profiles for MV infected cells at 3 days post-infection (3 DPI) for varying MOI. Bar heights represent the number of cells with an optical force index falling within the given bin range (bin width = 2). The dotted line represents the optical force index threshold set as the Radiance infection metric (OFI > 35 s^−1^) for this study. The percentage of the cell population with an OFI > 35 s^−1^ is shown on each plot. (**D**). Comparison of Radiance infection metric (percentage of cells with OFI > 35 s^−1^) at 3 days post-infection (3 DPI) for cells infected with MV at varying levels across two unique cell infectivity experiments (*n* = 3). (**E**). Correlation between Radiance infection metric (percentage of cells with OFI > 35 s^−1^) and MOI for cells infected with MV and harvested at 3 days post-infection (3 DPI). Each line represents a separate experiment with unique cell source and stock virus titer. R^2^= 0.994 (*n* = 3) for Experiment 1, and R^2^= 0.999 (*n* = 3) for Experiment 2.

**Table 1 vaccines-10-01589-t001:** Calculated Potency Values for MV-infected bioreactor samples. Table detailing days post-infection (DPI), Radiance infection metric, total virus particle counts used as surrogate potency, and calculated potency in terms of vp/viable cell based on the correlation between the LFC data and flow cytometry data for all three MV 3 L bioreactor experiments. The log_10_ difference between the measured and calculated potencies for each sample timepoint as well as the absolute average for all samples across all three experiments are given in the final column.

Experiment	DPI	Radiance Infection Metric (% OFI > 55 s^−1^)	CytoFLEX Surrogate Potency (vp/Viable Cell)	Calculated Potency (vp/Viable Cell)	Log_10_ Difference
Exp 1	3	11.53	3.15 × 10^3^	3.57 × 10^3^	0.055
	4	23.52	6.86 × 10^3^	5.87 × 10^3^	−0.068
	5	40.92	1.57 × 10^4^	1.20 × 10^4^	−0.115
Exp 2	3	9.49	3.17 × 10^3^	3.29 × 10^3^	0.015
	4	22.60	4.56 × 10^3^	5.65 × 10^3^	0.093
	5	42.57	8.73 × 10^3^	1.29 × 10^4^	0.169
	6	54.02	2.33 × 10^4^	2.07 × 10^4^	−0.052
Exp 3	3	28.12	7.05 × 10^3^	7.10 × 10^3^	0.003
	4	29.69	1.12 × 10^4^	7.57 × 10^3^	−0.169
	5	41.48	1.16 × 10^4^	1.23 × 10^4^	0.025
	6	65.92	3.05 × 10^4^	3.38 × 10^4^	0.045
				**Absolute Average**	**0.074**

**Table 2 vaccines-10-01589-t002:** Calculated Potency Values for AV1-infected bioreactor samples. Table detailing days post-infection (DPI), Radiance infection metric, total virus particle counts used as surrogate potency, and calculated potency in terms of vp/viable cell based on the correlation between the LFC data and flow cytometry data for all three AV1 3 L bioreactor experiments. The average log_10_ difference between the measured and calculated potencies for each sample timepoint as well as the absolute average for all samples across all three experiments are given in the final column.

Experiment	DPI	Radiance Infection Metric (% OFI > 55 s^−1^)	CytoFLEX Surrogate Potency (vp/Viable Cell)	Calculated Potency (vp/Viable Cell)	Log_10_ Difference
Exp 1	3	21.06	5.72 × 10^2^	5.29 × 10^2^	−0.034
	4	43.19	2.26 × 10^3^	1.68 × 10^3^	−0.129
	5	54.78	4.41 × 10^3^	3.09 × 10^3^	−0.155
Exp 2	3	19.72	4.95 × 10^2^	4.93 × 10^2^	−0.002
	4	35.46	9.71 × 10^2^	1.12 × 10^3^	0.064
	5	45.07	1.45 × 10^3^	1.86 × 10^3^	0.108
	6	72.13	6.80 × 10^3^	7.66 × 10^3^	0.052
Exp 3	3	34.24	4.96 × 10^2^	1.05 × 10^3^	0.328
	4	27.07	1.06 × 10^3^	7.25 × 10^2^	−0.166
	5	37.75	1.41 × 10^3^	1.27 × 10^3^	−0.046
	6	54.44	3.18 × 10^3^	3.04 × 10^3^	−0.020
				**Absolute Average**	**0.100**

**Table 3 vaccines-10-01589-t003:** Calculated Potency Values for MV Cell Infectivity Assay Samples. Table detailing the Radiance infection metric, Calculated MOI, and Calculated Titer based on the correlation between the LFC data and Input MOI for each cell infectivity experiment. Finally, the average log_10_ difference between the stock titer and calculated titer for each sample MOI as well as the average for all points in a given experiment are given in the final column.

Experiment 1					
Input MOI (TCID50/Cell)	Level (µL)	Radiance^®^ Infection Metric (% Cells with OFI > 35 s^−1^)	Calculated MOI (TCID50/Cell)	Calculated Titer (TCID50/mL)	Log_10_ Difference from Stock
Stock	N/A	N/A	N/A	8.20 × 10^6^	N/A
4.100	100	26.9	5.299	1.06 × 10^7^	0.111
1.640	40	20.1	1.694	8.47 × 10^6^	0.014
0.328	8.0	12.6	0.276	6.90 × 10^6^	−0.075
0.066	1.6	7.5	0.037	4.57 × 10^6^	−0.253
0.013	0.32	5.7	0.013	8.15 × 10^6^	−0.003
0.003	0.064	4.3	0.004	1.33 × 10^7^	0.211
			**Average**	**8.67 × 10^6^**	**0.024**
**Experiment 2**					
Stock	N/A	N/A	N/A	6.81 × 10^6^	N/A
3.407	100	44.3	3.227	6.45 × 10^6^	−0.024
1.363	40	37.3	1.405	7.03 × 10^6^	0.013
0.273	8.0	27.0	0.295	7.39 × 10^6^	0.035
0.055	1.6	18.7	0.051	6.33 × 10^6^	−0.032
0.011	0.32	13.7	0.011	6.91 × 10^6^	0.006
			**Average**	**6.82 × 10^6^**	**0.001**

## Data Availability

The data presented in this study are available on request from the corresponding author. The data are not publicly available due to the proprietary nature of data which were used that complies with the requirements of the current legal framework at Merck. Data pseudo-anonymized are however available from the MMD labs upon reasonable request to any researcher wishing to use them for non-commercial purposes and will have to be approved by Merck legal prior to sharing.

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
