# Peer review of "Rapid In-Process Measurement of Live Virus Vaccine Potency Using Laser Force Cytology: Paving the Way for Rapid Vaccine Development"

_vaccines, 2022, doi:10.3390/vaccines10101589_

Round 1
Reviewer 1 Report
​The current manuscript presents a method to perform some "surrogate potency measurement by looking at cellular “fingerprint” upon virus infection", with the focus on live-virus vaccines (LVVs). In the light of the current corona, monkeypox, etc. world-wide outbreaks this research is extremely timely and important. I clearly support publication, subject to some moderate [often cosmetic] revision, please see the points to be addressed being listed below.
In the introduction, the authors can add some concrete examples of LVVs which are/were used broadly. One should also mention if the method proposed in the manuscript would help to improve or direct the design of modern vaccines employed broadly. This way the importance of the current development and of the method of Laser Force Cytology (LFC) will be supported.
Towards the end of the introduction the authors should concisely but clearly state what they do, how they do it (methods, approaches, models, etc.), and what exactly do they get as the results. A sectioned plan of the entire paper is to be presented here.
In many figures the blue and red empty circles used as symbols will be hard to distinguish in a black-white printout: please use DIFFERENT symbols, in addition to different colors, systematically in all figures having this problem.
The same applies to the plots with interconnected symbols, as in e.g. Fig. 3D.
The main text lacks structure/sectioning: adding well-thought and descriptive subsections would be very beneficial here.
The process of virus entry into cells often involves binding and invagination of the entire virions or of their genetic material, such as the DNA. In this respect the authors are encouraged to describe this process---as one of the limiting steps of viral infection involving DNA-membrane binding for DNA-viruses or membrane-membrane fusion for ss-RNA viruses [such as enveloped HIV-1 or influenza viruses]---in the main text. This is often a limiting step of infection that can potentially be influenced by certain agents facilitating or inhibiting the cell-entry process.
Here the experimental study of virus diffusion and virus-virus on a membrane can be mentioned [DOI: 10.1039/C7SM00829E] as well as the theoretical description of DNA-membrane binding performed in Ref. [13] of the above mentioned study. Virus-cell binding was also a subject of a number of computer-simulation studies. I feel that a deeper description of key biophysical and physical aspects of the virus-cell entry process---affecting the efficiency of many vaccines possibly---can be beneficial for the revised version of the current manuscript.
I like the shift to covid in the discussion; also an interesting twist is usage of advantageous agents. Here, however, i think the authors can be more specific regarding classifying concretely which method brings some advantages for which viruses, ss-RNA versus ds-DNA for instance. For these two classes of viruses, i can imaging that e.g. that the mechanisms of entry of virions to host cells will be different and thus the respective strategies tuning this entry process i presume as well. From this perspective, the main text can be also restructured a little to show which types of viruses respond to these or those strategies of vaccination improvement.
Author Response
Reviewer 1
Comment 1 : The current manuscript presents a method to perform some "surrogate potency measurement by looking at cellular “fingerprint” upon virus infection", with the focus on live-virus vaccines (LVVs). In the light of the current corona, monkeypox, etc. world-wide outbreaks this research is extremely timely and important. I clearly support publication, subject to some moderate [often cosmetic] revision, please see the points to be addressed being listed below.
In the introduction, the authors can add some concrete examples of LVVs which are/were used broadly. One should also mention if the method proposed in the manuscript would help to improve or direct the design of modern vaccines employed broadly. This way the importance of the current development and of the method of Laser Force Cytology (LFC) will be supported.
Response 1: We would like to thank the reviewer for such a positive review of our manuscript and the support in favor of publishing it. We have expanded the introduction section based on these suggestions and have included a section that talks about application of the Lumacyte Radiance technology for other types of vaccines (virus like particles, mRNA vaccines, etc.). We have also talked about application of this method to older and already commercialized LVVs that might benefit from the use of such a tool. All added sections in the manuscript have been highlighted in a different color for easy identification.
Comment 2: Towards the end of the introduction the authors should concisely but clearly state what they do, how they do it (methods, approaches, models, etc.), and what exactly do they get as the results. A sectioned plan of the entire paper is to be presented here.
Response 2: We want to thank the reviewer again for this comment and have added a stepwise summary of the manuscript in the introduction. We agree that this will significantly improve the reader’s quality of understanding of this work in the latter part and in the results sections.
Comment 3: In many figures the blue and red empty circles used as symbols will be hard to distinguish in a black-white printout: please use DIFFERENT symbols, in addition to different colors, systematically in all figures having this problem.
Response 3: Figure symbols and colors have been adjusted in Figure 2, 3, 4 and 5 to improve clarity for grayscale printing
Comment 4: The same applies to the plots with interconnected symbols, as in e.g. Fig. 3D.
Response 4: Figure symbols and colors have been adjusted in Figure 2, 3, 4 and 5 to improve clarity for grayscale printing
Comment 5: The main text lacks structure/sectioning: adding well-thought and descriptive subsections would be very beneficial here.
Response 5: We have added descriptive subsections before each main figure in the manuscript. Once again, we agree that this significantly improves the reading quality of this paper and would like to thank the reviewer for this thoughtful comment.
Comment 6: The process of virus entry into cells often involves binding and invagination of the entire virions or of their genetic material, such as the DNA. In this respect the authors are encouraged to describe this process---as one of the limiting steps of viral infection involving DNA-membrane binding for DNA-viruses or membrane-membrane fusion for ss-RNA viruses [such as enveloped HIV-1 or influenza viruses]---in the main text. This is often a limiting step of infection that can potentially be influenced by certain agents facilitating or inhibiting the cell-entry process.
Here the experimental study of virus diffusion and virus-virus on a membrane can be mentioned [DOI: 10.1039/C7SM00829E] as well as the theoretical description of DNA-membrane binding performed in Ref. [13] of the above mentioned study. Virus-cell binding was also a subject of a number of computer-simulation studies. I feel that a deeper description of key biophysical and physical aspects of the virus-cell entry process---affecting the efficiency of many vaccines possibly---can be beneficial for the revised version of the current manuscript.
Response 6: Relevant text has been added in the discussion section to address the different stages of viral infection and how mechanistic studies using LFC could potentially help optimize these stages to improve vaccine design by reducing the turnaround time of bulk potency assays. The two references mentioned here have also been added to the manuscript.
Comment 7: I like the shift to covid in the discussion; also an interesting twist is usage of advantageous agents. Here, however, i think the authors can be more specific regarding classifying concretely which method brings some advantages for which viruses, ss-RNA versus ds-DNA for instance. For these two classes of viruses, i can imaging that e.g. that the mechanisms of entry of virions to host cells will be different and thus the respective strategies tuning this entry process i presume as well. From this perspective, the main text can be also restructured a little to show which types of viruses respond to these or those strategies of vaccination improvement.
Response 7: Thank you for the suggestion, some text has been added to the document to address this comment. In addition, though not included in this study, Lumacyte has collected data demonstrating the potential to classify and differentiate the cellular response as measured by Laser Force Cytology to different viruses. This work is yet un-published and hence no reference can be provided currently.
Reviewer 2 Report
The study by McCracken et al. described a novel assay (laser force cytology) to measure live virus vaccines for process analytical evaluations. Compared to traditional plaque or TCID50 assays, the LFC demonstrates high correlation with traditional assays, and rapid, robust capabilities. They validated this test by testing two different viruses. The study was designed logically and the overall manuscript was well-written and the topic is within journal’s scope. One minor comment I have is that please specific details of the additional virus (AV1) and the sources of both viruses as well.
Author Response
Reviewer 2
Comment 1: The study by McCracken et al. described a novel assay (laser force cytology) to measure live virus vaccines for process analytical evaluations. Compared to traditional plaque or TCID50 assays, the LFC demonstrates high correlation with traditional assays, and rapid, robust capabilities. They validated this test by testing two different viruses. The study was designed logically and the overall manuscript was well-written and the topic is within journal’s scope. One minor comment I have is that please specific details of the additional virus (AV1) and the sources of both viruses as well.
Response 1: First we want to thank the reviewer for this thoughtful comment and for recommending our manuscript for publication. Because of confidentiality agreements held within Merck and the fact that the AV1 virus used in our study is yet an unlicensed product, we are unable to reveal the name of this virus. We apologize for this inconvenience. However, we have added the genus of the AV1 virus (Flavivirus) used in this study and we hope that this clarifies any questions around the application of this as a proof of concept to co-detect two very similar enveloped viruses in a single bioreactor. We think that, since the Radiance technique is able to distinguish between two very similar entities (enveloped LVVs), it confirms a worst case scenario of detecting an unknown adventitious agent. In reality, we think that it will be far easier to detect a real adventitious agent coming from a completely different class of pathogen (bacteria, fungi, mycoplasma, etc.). A discussion around this has been added in the discussion section of the manuscript.